# Challenges and Limitations of Current RSV Prevention Strategies in Infants and Young Children: A Narrative Review

**DOI:** 10.3390/vaccines13070717

**Published:** 2025-07-01

**Authors:** Nicola Principi, Serafina Perrone, Susanna Esposito

**Affiliations:** 1Università degli Studi di Milano, 20122 Milan, Italy; nicola.principi@unimi.it; 2Neonatology Unit, Department of Medicine and Surgery, University of Parma, 43126 Parma, Italy; serafina.perrone@unipr.it; 3Pediatric Clinic, Department of Medicine and Surgery, University of Parma, 43126 Parma, Italy

**Keywords:** respiratory syncytial virus, RSV, nirsevimab, maternal RSV vaccination, RSV preF vaccine, monoclonal antibodies

## Abstract

*Background:* Respiratory syncytial virus (RSV) remains a leading cause of lower respiratory tract infections and hospitalizations in infants and young children globally. Recently, RSV prevention has advanced with the introduction of nirsevimab, a long-acting monoclonal antibody, and the RSV preF vaccine for maternal immunization. While these interventions have improved early protection, several limitations hinder their broader impact and long-term effectiveness. *Methods:* This narrative review synthesizes evidence from clinical trials, observational studies, and regulatory reports to evaluate the main limitations of nirsevimab and maternal RSV vaccination. Literature searches were conducted in major databases, focusing on efficacy, safety, immunogenicity, implementation, and population-specific challenges. *Results*: Both nirsevimab and maternal vaccination provide strong protection during the first six months of life, but their effectiveness wanes thereafter. This is concerning as nearly half of RSV-related deaths occur in children over six months old. Maternal vaccine efficacy is uncertain in very-preterm infants, and safety concerns persist, including potential associations with preterm birth, Guillain–Barré syndrome, and hypertensive disorders. Real-world data from low-income countries are lacking, limiting generalizability. Additionally, the risk of vaccine-associated enhanced disease (VAED), although unconfirmed, has delayed pediatric vaccine development. Emerging monoclonal antibodies and live-attenuated vaccines are under investigation to extend protection beyond infancy. *Conclusions*: Despite substantial progress, current RSV prevention strategies leave critical gaps, particularly for older infants and underserved populations. There is a pressing need for next-generation vaccines, enhanced pharmacovigilance, and equitable global implementation to ensure sustained and inclusive RSV protection.

## 1. Introduction

Respiratory syncytial virus (RSV) was first identified more than 50 years ago, alongside growing awareness of its substantial medical, social, and economic impact on the pediatric population [1,2]. Figure 1 shows RSV symptoms according to age group, whereas Figure 2 describes its burden in the first year of life.

Despite this early recognition, effective preventive measures were not developed until decades later, primarily due to the failure of a formalin-inactivated vaccine in the 1960s. This vaccine was not only ineffective but also harmful: its administration in RSV-naive children led to the development of vaccine-associated enhanced disease (VAED) [3], characterized by severe lower respiratory tract infections (LRTIs) following subsequent exposure to the natural virus [4].

For nearly two decades, this adverse outcome hampered further efforts to develop pharmacological preventions for RSV. As a result, the global burden of RSV remained unchanged until the late 1990s. Although human intravenous immunoglobulins (IVIGs) and specialized preparations such as RespiGam (IVIG enriched with high titers of anti-RSV antibodies) were introduced, their impact on the overall burden of RSV was limited.

Significant progress was achieved only towards the end of the 20th century, when deeper insights into RSV structure and antigenicity enabled the development of more-effective preventive strategies. It was identified that the fusion (F) and attachment (G) proteins of RSV play central roles in viral infectivity and in eliciting protective antibody responses. In particular, the F protein exists in two conformational states—prefusion (pre-F) and postfusion (post-F). The pre-F state contains several potent neutralizing epitopes (sites Ø and I–V), while the post-F state presents sites I–IV [5,6,7]. These discoveries paved the way for novel vaccines and monoclonal antibodies (mAbs) designed to target these key epitopes.

Recently, two major innovations have been introduced: nirsevimab, a monoclonal antibody that targets site Ø on the pre-F protein, and the RSV preF vaccine, a bivalent, unadjuvanted formulation containing preF A and B proteins intended for use in pregnant women. Nirsevimab is recommended for all neonates and young infants entering their first RSV season, as well as high-risk children prior to their second season. A single dose has demonstrated sustained efficacy for at least 180 days, preventing up to 80% of RSV-related hospitalizations and over 75% of severe LRTIs [8,9]. Meanwhile, maternal administration of the RSV preF vaccine during pregnancy allows for the passive protection of infants via the transplacental transfer of neutralizing antibodies, significantly reducing RSV-related hospitalizations in the first 3–6 months of life [10,11].

Despite the substantial public health gains associated with these preventive tools, concerns remain. Several limitations may compromise their long-term efficacy or applicability to specific subgroups, including preterm infants, children with comorbidities, or populations in low-income settings. Understanding these limitations is crucial for optimizing RSV prevention strategies.

This is a narrative review aimed at exploring and discussing the main limitations of nirsevimab and the RSV preF maternal vaccine. The review emphasizes the clinical relevance of each limitation and outlines possible solutions or alternative strategies.

## 2. Methods

To perform this review, we conducted a literature search using PubMed, Scopus, Web of Science, and the Cochrane Library. Search terms included combinations of the following keywords: “respiratory syncytial virus,” “RSV,” “Nirsevimab,” “maternal RSV vaccine,” “RSV preF,” “vaccine-associated enhanced disease,” “VAED,” “resistance,” “antidrug antibodies,” “infants,” and “children.” Boolean operators (AND, OR) were applied to refine the search. The search included publications from inception through to April 2025, with no language restrictions. Preference was given to peer-reviewed articles, randomized controlled trials, systematic reviews, and official reports or guidelines from organizations such as the WHO, CDC, EMA, and FDA. References from relevant articles were also screened manually to identify additional pertinent studies. We included studies that addressed RSV prophylaxis with monoclonal antibodies or maternal vaccines; discussed safety, immunogenicity, resistance, or efficacy in pediatric populations; and explored real-world limitations or implementation challenges. Exclusion criteria included studies limited to adult or elderly populations and narrative pieces lacking original data or systematic methodologies.

## 3. Nirsevimab: Potential Limitations

### 3.1. Emergence of Nirsevimab-Resistant RSV Strains

The RSV F protein is highly conserved among viral strains, and, therefore, neutralizing antibody preparations targeting major antigenic sites of this protein are expected to retain long-term efficacy. This is the case with nirsevimab, which binds to antigenic site Ø on the prefusion (pre-F) form of the RSV F protein, thereby blocking viral entry into host cells. However, due to the genetic variability of RSV [12], there is a theoretical risk of emerging variants with mutations that confer reduced susceptibility to nirsevimab, particularly under selective pressure from widespread prophylactic use.

Fortunately, naturally occurring nirsevimab-resistant RSV strains have been extremely rare across different regions and time periods, as demonstrated by several studies that screened circulating viruses for mutations in the nirsevimab binding site [13,14,15]. Moreover, post-marketing surveillance and clinical trials conducted after nirsevimab introduction have confirmed that the emergence of resistance remains very limited and primarily involves RSV B strains.

In two Phase IIb/III clinical trials evaluating nirsevimab efficacy and viral genetic profiles in breakthrough infections [16], more than 99% of isolates remained susceptible to nirsevimab, with no significant resistance mutations detected in RSV A strains. Although several substitutions were identified in RSV B strains, most were actually associated with increased nirsevimab susceptibility. Only two participants (1%) exhibited an isolate with triple substitutions in site Ø that conferred resistance. Similar findings were reported by Fourati et al. [17] in a study involving 695 RSV-infected infants, 349 of whom had received nirsevimab. Among 472 RSV A strains analyzed (half from treated infants), no resistance-associated mutations were found. In 2 of the 24 RSV B breakthrough cases, mutations linked to nirsevimab resistance were identified—F:N208D (previously described in [18]) and a novel combination (F:I64M + F:K65R).

While no nirsevimab resistance has been detected in RSV A strains, the emergence of resistant RSV B variants underlines the potential risk of reduced nirsevimab efficacy in the future. Nonetheless, current data indicate that this risk is extremely low and does not warrant the replacement of nirsevimab with alternative measures. This is further supported by epidemiological data showing that RSV A strains are generally more prevalent than RSV B strains across most regions and seasons [19].

To mitigate this risk, molecular surveillance is essential to detect and track resistant variants and to assess the role of nirsevimab-induced selective pressure. Meanwhile, the development of alternative monoclonal antibodies targeting different epitopes should continue.

### 3.2. New Monoclonal Antibodies Potentially Effective Against Nirsevimab-Resistant Strains

Clesrovimab, a long-acting monoclonal antibody targeting site IV of the RSV F protein, has shown promising results. Clinical trials in healthy preterm and term infants have demonstrated that clesrovimab reduced RSV-related acute respiratory infections by 50%, hospitalizations by 81.3%, and severe LRTIs by 91.7% [20] It can be administered using the same dose regardless of the child weight and has been recently approved for use in infants born during or entering their first RSV season [21].

Another potential candidate is 5B11, an antibody targeting a highly conserved, immunologically tolerant epitope within site V of the RSV F protein. In vitro, 5B11 demonstrated potent, balanced neutralization against both RSV A and B strains, including nirsevimab-resistant variants like B18537 [22]. In animal models, 5B11 outperformed nirsevimab in preventing infection [23]. However, further studies are needed to evaluate its safety, tolerability, and efficacy in children before it can be considered for clinical use.

Similar considerations apply to TNM001 (trinomab), which targets the RSV pre-F protein and is currently being assessed in a Phase III clinical trial (NCT06083623) involving preterm and term infants during their first RSV season.

### 3.3. Production of Antidrug Antibodies and Increased Antibody Clearance

Some children receiving nirsevimab have been found to develop antidrug antibodies (ADAs). While ADAs have shown minimal effects on nirsevimab pharmacokinetics and no significant impact on safety, their clinical relevance warrants further investigation. At 151 days post-administration, neutralizing antibody concentrations in ADA-positive children were comparable to levels associated with protection against severe RSV infection [24]. However, approximately one year after immunization, NS levels in ADA-positive children were lower than in those without ADAs [24]. This observation suggests that ADAs may reduce the efficacy of a second nirsevimab dose, which is recommended for high-risk children prior to their second RSV season.

In addition, increased nirsevimab clearance has been reported in a small number of children with underlying conditions that may cause protein loss. In these cases, nirsevimab serum concentrations were lower than in healthy controls, although still within a range likely to support protective efficacy [25]. However, due to the limited number of subjects studied, definitive conclusions cannot yet be drawn, and the influence of specific underlying conditions remains unclear.

### 3.4. Development of Vaccine-Associated Enhanced Disease (VAED)

To extend nirsevimab’s half-life and provide season-long protection with a single dose, the antibody was engineered with a triple amino acid substitution (M252Y/S254T/T256E) in its Fc region. This modification enhances binding to the neonatal Fc receptor (FcRn), increasing recycling and preventing degradation. Consequently, the half-life of nirsevimab in healthy late-preterm and term infants has been extended from 21–28 days to approximately 70 days, enabling effective prophylaxis throughout a full RSV season with one intramuscular injection [26].

Clinical trials and post-marketing studies have confirmed that nirsevimab is generally safe and well tolerated [27]. Nonetheless, some concerns have been raised regarding its potential to cause vaccine-associated enhanced disease (VAED). Banoun [28], reviewing clinical and preclinical data, argued that, while rare, NS might theoretically exacerbate RSV infection in some cases through VAED mechanisms. These mechanisms involve interactions between the Fc region of antibodies and cellular receptors or the complement system, potentially facilitating viral entry and inflammation.

Although nirsevimab primarily acts through direct viral neutralization, its Fc fragment may still mediate other effector functions, such as antibody-dependent neutrophil phagocytosis (ADNP), cellular phagocytosis (ADCP), complement deposition (ADCD), and cellular cytotoxicity (ADCC), which could influence outcomes [29]. At sub-neutralizing concentrations, these mechanisms might theoretically promote viral entry and inflammation, leading to severe disease [30].

Support for this concern includes clinical observations suggesting more-severe RSV cases in immunized children, as well as a temporal association between nirsevimab immunization campaigns and a notable increase in neonatal mortality within the 2–6-day age group in France, particularly during the period of RSV circulation. However, early VAED cases might have gone undetected, as some post-marketing studies excluded neonates hospitalized shortly after vaccination, when nirsevimab concentrations might still be below protective levels [31].

Nevertheless, no VAED cases have been reported in preclinical, clinical, or post-marketing studies involving nirsevimab. Laboratory comparisons of nirsevimab and palivizumab revealed similar levels of Fc effector activity (ADNP, ADCP, and ADCD), and lower ADCC activity for nirsevimab (yet still significantly above the baseline). Palivizumab has been used for over 25 years with an excellent safety record and no known association with VAED [32,33]. Given the similarity in Fc-mediated effects between nirsevimab and palivizumab, the risk of VAED with nirsevimab appears to be minimal.

Despite this reassuring data, ongoing pharmacovigilance and close monitoring of immunized children, particularly during their first and second RSV seasons, remain essential to definitively exclude the occurrence of VADE-related complications.

## 4. RSV Vaccine in Pregnant Women

### 4.1. Limitations in Current Evidence and Implementation

Several limitations exist in the current body of evidence regarding the effectiveness of the RSV preF vaccine when administered to pregnant women, particularly in reducing RSV infections in neonates and young infants. These evidence gaps highlight the need for specifically designed studies. For example, the vaccine’s efficacy in preterm infants remains uncertain, largely because very few children born before 32 weeks of gestation were included in clinical trials [10].

Furthermore, available data primarily reflect outcomes in pregnant women without underlying risk factors. Women with conditions affecting immune function, a history of preterm birth, multiple pregnancies, or previous children with significant congenital anomalies were generally excluded from studies. Understanding the vaccine’s effectiveness in these higher-risk groups is crucial, as it may inform the need for additional or alternative preventive strategies. In cases where efficacy is reduced, combined approaches, such as maternal vaccination plus postnatal administration of nirsevimab, may be advisable.

Another limitation concerns geographical and socioeconomic representation. Most clinical trials of the RSV preF vaccine have been conducted in high- and middle-income countries, with limited data from low-income settings. Given the higher burden of severe RSV disease in these regions [34], maternal vaccination may prove even more beneficial. However, the effectiveness of maternal RSV vaccination may vary across different populations due to factors like pre-existing immunity, nutritional status, and the prevalence of other health conditions. Moreover, ensuring equitable access to vaccination services, particularly for women in remote or underserved areas, is crucial. This may involve addressing geographical barriers, cultural beliefs, and socioeconomic factors that influence healthcare utilization. Additional studies focused on low- and middle-income countries are essential to assess the vaccine’s global applicability and support its broader implementation.

Finally, maternal vaccination rates remain generally low, and uptake is closely tied to perceptions of RSV severity. Therefore, educational initiatives targeting healthcare professionals (particularly obstetricians, midwives, and primary care providers) are needed to promote routine antenatal RSV vaccination worldwide [35].

### 4.2. RSV preF Vaccine Safety

Although the RSV preF vaccine is authorized for use in pregnant women, several safety concerns remain unresolved. Chief among them is a possible association with an increased risk of preterm birth. In one study, women vaccinated between 24 and 36 weeks of gestation had a higher rate of preterm birth compared to those receiving a placebo (5.7% vs. 4.7%, respectively; relative risk [RR] 1.20, 95% CI 0.98–1.46) [36]. However, a definitive causal relationship could not be established, as other confounding factors may have influenced outcomes.

This analysis was limited by a low number of events, and the increased risk of preterm birth was observed primarily in non-high-income countries (RR 1.73, 95% CI 1.22–2.47), especially South Africa (RR 2.06, 95% CI 1.21–3.51). Notably, many premature births occurred during a period when the B.1.617.2 (Delta) variant of COVID-19 was predominant [36].

A small but statistically significant increase in preterm birth risk was also identified in a separate trial of the RSV preF3 vaccine, a formulation similar to the RSV preF vaccine. This study was terminated early after interim analysis showed a higher rate of preterm births among vaccinated women (6.8% vs. 4.9%; RR 1.37; 95% CI 1.08–1.74; *p* = 0.01). Neonatal mortality was also slightly higher in the vaccine group (0.4% vs. 0.2%; RR 2.16; 95% CI 0.62–7.56; *p* = 0.23) [37].

Due to these mixed findings, a Cochrane review of six clinical trials concluded that maternal RSV vaccination might slightly increase the risk of preterm birth, although the evidence was of very low certainty (RR 1.16; 95% CI 0.99–1.36; six RCTs, 17,560 infants). These safety concerns have influenced regulatory guidance: In the United States, the vaccine is recommended between 32 and 36 weeks of gestation to minimize any potential risk. In contrast, in Europe and the UK, vaccination is permitted from 24 to 36 weeks. Clarifying these safety issues is vital, as widening the immunization window may improve vaccine coverage, particularly in low-resource settings where prenatal care is often limited.

Another unresolved concern involves the potential association between RSV preF vaccination and Guillain–Barré syndrome (GBS). Among adults aged 65 and older, nine additional GBS cases per million doses were reported within 1 to 42 days post-vaccination. Although the FDA could not establish a causal link between the vaccine and GBS, it did not rule out the possibility, especially in pregnant women. Consequently, the agency has requested further safety studies and mandated updates to the product labeling to inform recipients of this potential risk [38].

Finally, additional research is needed to evaluate whether RSV preF vaccination is associated with early- or late-onset hypertensive disorders of pregnancy. A recent analysis reported a hazard ratio (HR) of 1.43 (95% CI 1.16–1.77) for developing hypertensive disorders following vaccination [39]. It remains unclear whether cases of early-onset hypertension—which would occur before vaccine eligibility—were excluded from the analysis. Furthermore, results were not stratified by the type of labor onset, making it difficult to determine whether spontaneous preterm births differed by vaccination status. The uncertainty surrounding the observed safety signals after RSV vaccination emphasizes the need for diligent post-marketing surveillance and the careful monitoring of reported adverse events following immunization. In this context, voluntarily submitted reports to the U.S. Food and Drug Administration’s Vaccine Adverse Event Reporting System (VAERS) can be important, particularly for evaluating rare but clinically relevant adverse events [40].

## 5. Protection of Older Infants and Children at Risk

The RSV preF vaccine is not authorized for use in infants due to the potential risk of VAED. Moreover, the protection conferred by both maternal immunization and nirsevimab administration diminishes over time. As a result, children older than six months receive little to no protection against RSV infection from these measures [39].

Although the majority of severe RSV infections occur within the first six months of life, RSV can still cause hospitalization and death in children up to five years of age. In 2019, it was estimated that the global hospitalization rate for RSV infection among children aged 6 to 12 months was approximately 10%. Furthermore, 44% of RSV-related deaths in children under five occurred in those older than six months [40]. In addition, prematurity and underlying comorbidities remain significant risk factors for severe RSV disease well beyond infancy, up to five years of age [41]. These findings highlight the urgent need for preventive strategies that extend protection to older infants and preschool-aged children.

To address this gap, several types of RSV vaccines have been developed, including live-attenuated, chimeric, vector-based, and mRNA vaccines. An ideal RSV vaccine for infants should elicit strong neutralizing antibody responses, robust T-helper cell activity, and sufficient CD8+ T-cell responses to ensure viral clearance. Among these, live-attenuated vaccines are particularly promising because they can induce broad innate, humoral, and cellular immune responses without the risk of VAED [42,43]. Additionally, they are administered as nasal drops, which are less invasive and better tolerated in young children. Importantly, these vaccines can also generate IgA-mediated mucosal immunity, helping to block viral entry at the respiratory mucosa [44].

To achieve an optimal balance between attenuation and immunogenicity, current strategies focus on knocking out or modifying specific RSV genes, such as M2-2, NS2, SH, and G [45,46]. Among the most-studied candidates are two vaccines with deletions in the NS2 gene: RSV ΔNS2/Δ1313/I1314L and RSV/6120/ΔNS2/1030s. Both have demonstrated safety and immunogenicity, though RSV/6120/ΔNS2/1030s exhibits higher replication, suggesting potentially greater immunogenicity, but also a higher incidence of mild adverse events, primarily rhinorrhea [47].

A Phase 3 clinical trial is planned to assess the efficacy of these live-attenuated vaccines in children aged 6 to 21 months compared to a placebo (NCT06252285), though recruitment has not yet begun.

In contrast, chimeric and vector-based RSV vaccines initially proposed for pediatric use have not progressed sufficiently to reach clinical application in children [48,49,50,51,52]. Currently, the U.S. Food and Drug Administration (FDA) has paused all RSV vaccine trials involving infants and RSV-naive children aged 2–5 years, pending further safety evaluation [48].

This decision was prompted by the results of a clinical trial evaluating two mRNA vaccines encoding the RSV preF protein, targeting either RSV alone or RSV combined with human metapneumovirus. In that study, involving 40 children aged 5 to 7 months, 5 (12.5%) of the vaccinated children who contracted RSV developed severe lower respiratory tract infections, compared to 1 out of 20 controls (5%) [49]. While the small sample size limits the ability to draw definitive conclusions, the findings raised concern over the potential for VAED, necessitating caution in the development of pediatric RSV vaccines.

## 6. Conclusions

Table 1 summarizes the timeline of key milestones in RSV vaccine and mAb development.

While some concerns regarding the use of nirsevimab and the RSV preF vaccine remain, current evidence suggests that these issues are generally of limited clinical significance and do not justify changes to existing recommendations despite clarification on supposed rare safety concerns being needed. Key areas requiring further clarification include the effectiveness of maternal immunization in very-premature infants and the potential benefits of combining maternal vaccination with the postnatal administration of nirsevimab.

A major limitation, however, is the reduced or absent protection offered by both nirsevimab and maternal immunization in children older than six months—even though RSV continues to pose a significant clinical risk up to five years of age. Although several candidate vaccines are under development to address this gap, including live attenuated and mRNA vaccines targeting RSV alone or the combination of RSV and other respiratory viruses, none have yet reached a level of maturity that permits clinical use. In particular, unresolved concerns regarding VAED remain a major barrier, as highlighted by the recent FDA decision to halt pediatric RSV vaccine trials pending further safety data.

Moreover, it is important to acknowledge that most of the available data on the efficacy, safety, and acceptability of these preventive interventions have been generated in high-income countries. There is a significant lack of evidence regarding their implementation and real-world impact in low-income settings, where logistical and economic barriers may hinder widespread use. This is particularly concerning given that the majority of RSV-related deaths occur in these regions. Therefore, comprehensive evaluations in low-resource environments are essential to fully understand and optimize the global utility of NS and the RSV preF vaccine.

## Figures and Tables

**Figure 1 vaccines-13-00717-f001:**
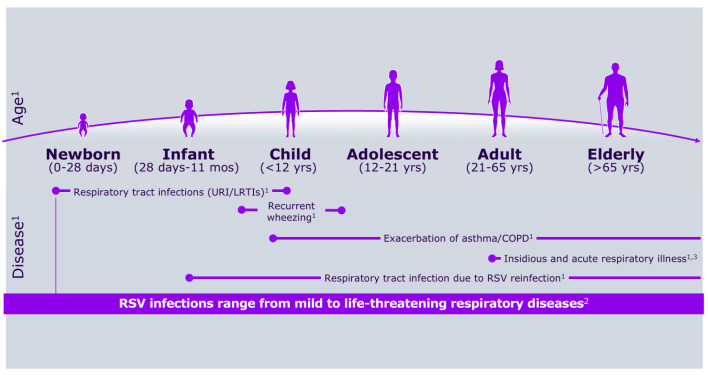
RSV symptoms according to age group.

**Figure 2 vaccines-13-00717-f002:**
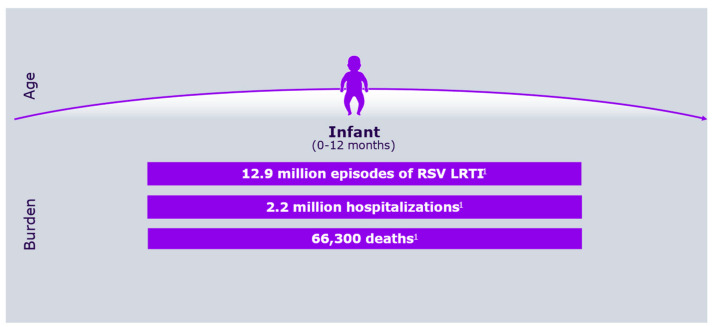
RSV overall burden in the first year of life.

**Table 1 vaccines-13-00717-t001:** Timeline of key milestones in RSV vaccine and monoclonal antibody development.

Year	Milestone
1956	RSV first identified in chimpanzees, later recognized in humans
1960s	Formalin-inactivated RSV vaccine trialed; associated with VAED, causing severe disease in infants
1990s	Introduction of RSV immune globulin (RespiGam); limited by intravenous administration
1998	Palivizumab (Synagis), the first monoclonal antibody for high-risk infants, approved
2010s	Structural studies of RSV F protein reveal preF and postF conformations; preF identified as key immunogenic target
2017	Nirsevimab engineered with Fc modifications for extended half-life
2022–2023	Phase III trials show strong efficacy of nirsevimab and RSV preF vaccine in infants and pregnant women
2023	Nirsevimab approved for all infants entering RSV season; RSV preF (maternal) vaccine approved in US and EU
2024–2025	Concerns raised over maternal vaccine safety; live-attenuated RSV vaccines enter advanced clinical trials
Ongoing	FDA halts pediatric RSV vaccine trials due to safety concerns (e.g., potential VAED); new candidates (e.g., clesrovimab, 5B11) in development

## Data Availability

Not applicable.

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
