# Peer review of "Challenges and Limitations of Current RSV Prevention Strategies in Infants and Young Children: A Narrative Review"

_vaccines, 2025, doi:10.3390/vaccines13070717_

Round 1

Reviewer 1 Report

Comments and Suggestions for Authors

Dear authors,

the manuscript is of great interest in the context of the recently approved vaccine.

You discuss all the relative concerns with an attractive approach.

However, due to the warning, already included in the vaccine about neurological issues, should be important to emphasize the value of the post approval pharmacovigilance ( VAERS system has reveiled the Guillain Barre cases ). Therefore a paragraph about passive, through the pharmacovigilance centers, and active ( registries of pregnant women, epidemiological studies) should be added in order the readers to be aware about the notification of safety signals.

Author Response

Thank you for your positive evaluation. We revised the manuscript according to your comment and those received from the other reviewers. 

A sentence concerning the relevance of VAERS system for detection of rare although clinically relevant adverse events has been added.

Reviewer 2 Report

Comments and Suggestions for Authors

The manuscript by Nicola Principi et al., is a narrative review on the latest advancements in respiratory syncytial virus (RSV) prevention. The core content is summarized as follows: RSV is a primary cause of lower respiratory infections in infants. In recent years, the introduction of the long-acting monoclonal antibody Nirsevimab and the RSV preF vaccine for maternal immunization represents significant progress in RSV prevention. Despite their effectiveness, these strategies have notable shortcomings. They provide robust protection for infants during the early months (up to 6 months), but the protective effect diminishes significantly after 6 months. This shortening of the protective window is a serious issue, as a significant proportion of RSV-related hospitalizations and deaths occur in children older than 6 months. In summary, the Principi team highlights that while Nirsevimab and maternal RSV vaccines are major breakthroughs, their short protection period (diminishing effectiveness after 6 months) fails to cover the high-risk population of older infants. Additionally, challenges such as unclear efficacy in premature infants, potential safety concerns, lack of data in low-income countries, and concerns related to the VAED (vaccine-associated enhanced disease) theory, necessitate the development of next-generation vaccines with longer-lasting protection. Addressing global health equity issues should also be prioritized. However, several shortcomings in the study need to be addressed to improve clarity and substantiate the conclusions drawn.

Major issues:

  1. The final paragraph of the Introduction (lines 67-82) provides a detailed description of the systematic literature search methodology (databases, keywords, screening process, inclusion/exclusion criteria). This entirely exceeds the scope of an "Introduction," making it overly lengthy and diluting its core purpose (introducing the background, problem, and research significance) while disrupting the logical structure of the paper.

  1. The core objective statement of the Introduction is: “This is a narrative review aimed at exploring and discussing the main limitations of current RSV preventive interventions, with a focus on nirsevimab and the RSV preF maternal vaccine.” (lines 67-69). While it clarifies the topic (limitations) and focus (the two products), it remains insufficient: the term "current" is ambiguous—does it include earlier interventions (e.g., palivizumab), or only the latest (nirsevimab and maternal vaccine)? The context suggests the latter, but this needs to be made more explicit.

  1. The core theme of Section 2.1 is "risk of resistance." The first half (lines 86-120) effectively discusses the theoretical basis of resistance, current status, clinical evidence, risk assessment (low), and mitigation strategies (monitoring). However, when introducing mitigation strategies such as "developing alternative antibodies" (line 120), the text suddenly devotes three paragraphs (lines 121-133) to detailing the mechanisms, data, and development stages of three alternative antibodies (clesrovimab, 5B11, TNM001). This severely deviates from the core theme of Section 2.1 ("risk of resistance"), over-expanding the discussion of "solutions" and overshadowing the main point. This makes Section 2.1 unnecessarily lengthy and blurs its focus, weakening the core message about the current status and risks of resistance. The detailed descriptions of these alternative antibodies would be better suited for an independent subsection on "future directions" or "alternative strategies," or as part of the Discussion section, rather than being nested within the "risk of resistance" subsection.

  1. In discussing concerns about VAED (lines 173-174), the authors state: “...as well as a temporal association between nirsevimab immunization campaigns and increased neonatal mortality in France.” This sentence only mentions a "temporal association" without providing any background information, data sources, investigation results, or official statements. This could easily lead to misinterpretation, implying a potential causal relationship (even if unintended by the authors) without supporting evidence. Strongly recommend deleting this sentence.

  1. The title of Subsection 3.1 is "Vaccine Use and Efficacy," but the content primarily discusses evidence gaps and applicability/accessibility issues rather than the actual "use" of the vaccine or direct reporting of its "efficacy." In reality, it discusses "what we still don’t know or need more research on." The title fails to accurately reflect the content’s focus, potentially misleading readers. Suggest revising the title to better align with the content. For example: “3.1. Evidence Gaps and Applicability Concerns” or “3.1. Limitations in Current Evidence and Implementation.”

  1. Section 4.1 significantly deviates from its core theme, which should be "existing protection gaps" and "exploration and challenges of solutions." Instead, most of the section (lines 270-299) focuses on detailing the technical aspects of live-attenuated vaccines and explaining the FDA pause. Recommend substantial trimming and restructuring of this section.

  1. Lines 314-320: “Moreover, it is important to acknowledge that most of the available data... in low-income settings... Therefore, comprehensive evaluations... are essential...” This discussion about the lack of data from low- and middle-income countries (LMICs) appears for the first time in the Conclusion. While Subsection 3.1 briefly mentions insufficient trial data from LMICs, the Conclusion elevates this to a standalone, critical "major limitation" linked to the "majority of RSV-related deaths." Introducing a new, significant argument (LMIC data gaps as a major limitation) in the Conclusion violates standard writing conventions. Conclusions should summarize and elevate points already thoroughly discussed in the main text. Recommend expanding the discussion of LMIC data/implementation gaps and their relationship to global disease burden in the relevant sections (e.g., Subsection 3.1 or a dedicated section on implementation challenges).

Minor Issues:

  1. “vaccine-associated enhanced disease (VAED)” (line 35) is correctly defined upon first use, but later the text abruptly uses “ADE” (line 73). VAED is the term specific to RSV literature referring to this historical event, while ADE (Antibody-Dependent Enhancement) is a broader immunological concept. Though related, VAED is more precise in the RSV context to avoid confusion.
  2. Line 87: “preparations containing neutralizing antibodies that target...” is slightly verbose. Suggest simplifying to: “neutralizing antibody preparations targeting...”.
  3. Lines 91-92: “emerging variants with mutations that confer reduced susceptibility to NS.” Recommend replacing “NS” with the full term “nirsevimab” or defining “NS” upon first use in the subsection (though it was used earlier, clarity within the subsection is better). Alternatively, consistently use “nirsevimab” (as done later).
  4. Line 102: “RSV A strains” and line 103: “RSV B isolates”—recommend standardizing terminology, using either “strains” or “isolates” throughout. While interchangeable in scientific writing, consistency is preferable.
  5. Line 104: “exhibited three substitutions”—is this referring to three mutations in a single viral strain or three different mutations across samples? The phrasing is slightly ambiguous. If the former, clarify as “a single isolate harboring three substitutions” or “an isolate with triple substitutions.” If the latter, rephrase as “exhibited mutations at three sites.”
  6. Lines 192-193: “These gaps highlight the need for specifically designed studies.” Suggest revising to: “These evidence gaps highlight the need for specifically designed studies.” (more fluid).

Author Response

Thank you for your suggestions. We revised the manuscript as recommended.

  1. All the sentences regarding literature search methodology have been moved to a specific paragraph.
  2. It has been clarified that the term current was refereed to new licensed measures for RSV prevention in children.
  3. All the data concerning new monoclonal antibodies for RSV prevention in neonates and young infants have been postponed to form a specific paragraph as suggested.
  4. The information regarding the temporal correlation between nirsevimab use and development of severe RSV infections in neonates when monoclonal antibody concentrations are still low seems important to highlight the need for continuous monitoring of RSV safety.
  5. The title has been modified as suggested.
  6. The section has been modified according to the suggestions.
  7. Several sentences to explain why further studies in LMIC are urgently needed and how they should be performed have been added.

      Minor issues. The text has been modified according all the suggestions.

Reviewer 3 Report

Comments and Suggestions for Authors

The paper of Nicola et al. is a narrative review about challenges and limitations of current RSV prevention strategies in infants and young children. Overall, the mansuscirpt was written in good English and summarized the main findings in this field.  However, some reviesions were still needed to improve the manuscript before consideration of publication.

Another figures and tables are still needed to increase readability of this paper for wide attention.

  1. Add a figure to clarify the development history of RSV vaccine.
  2. Add a figure in the introduction rate about the global burden of RSV.
  3. P67 to 84, the last paragraph in introduction is about the source of reference of this review. It would be better to put this paragraph in another section alone but not in introduction.
  4. In the conclusion part, the auhthors should give their opinion about the future development of RSV vaccine.

Author Response

Thank you for your suggestions. We revised the manuscript according to your comments.

  1. Two Figures have been added in the Introduction according to the suggestion.
  2. A Table has been added in the Conclusion considering your request.
  3. The methods used to select publications have been reported in a special section.
  4. In the Conclusions, data regarding vaccines presently in development have been added.

Round 2

Reviewer 3 Report

Comments and Suggestions for Authors

The revised manuscript has been improved. Accept in the current form